# Spatialtemporal evolution characteristics of ozone in China and its response to urbanization

**Li-Min Wang**[1,2], **Zi-Yi Ran**[3], **Xiang-Li Wu**[1,2]*, **Heng-Yu Wang**[3], **Li-Bin Zhao**[3]

**1** College of Geographical Sciences, Harbin Normal University, Harbin, China, **2** Key Laboratory of Remote Sensing Monitoring of Geographic Environment of Heilongjiang Province, Harbin Normal University, Harbin, China, **3** Artificial Rainfall Office of the People's Government of Heilongjiang Province, Harbin, China

* jndxwxl@163.com

**Data Availability Statement:** All relevant data are within the manuscript and its Supporting Information files.

**Funding:** This work was supported by the National Social Science Foundation of China (grant

## Abstract

Based on the background of urbanization in China, we used the dynamic spatial panel Durbin model to study the driving mechanism of ozone pollution empirically. We also analyzed the spatial distribution of ozone driving factors using the GTWR. The results show that: i) The average annual increase of ozone concentration in ambient air in China from 2015 to 2019 was $1.68 \mu g/m^3$, and $8.39 \mu g/m^3$ elevated the year 2019 compared with 2015. ii) The Moran's I value of ozone in ambient air was 0.027 in 2015 and 0.209 in 2019, showing the spatial distribution characteristics of "east heavy and west light" and "south low and north high". iii) Per capita GDP industrial structure, population density, land expansion, and urbanization rate have significant spillover effects on ozone concentration, and the regional spillover effect is greater than the local effect. R&D intensity and education level have a significant negative impact on ozone concentration. iv) There is a decreasing trend in the inhibitory effect of educational attainment and R&D intensity on ozone concentration, and an increasing trend in the promotional effect of population urbanization rate, land expansion, and economic development on ozone concentration. Empirical results suggest a twofold policy meaning: i) to explore the causes behind the distribution of ozone from the new perspective of urbanization, and to further the atmospheric environmental protection system and ii) to eliminate the adverse impacts of ozone pollution on nature and harmonious social development.

## Introduction

In the early days, China's economic development was driven at the expense of serious environmental pollution. With the intensification of atmospheric environmental pollution, strengthening environmental governance has become a rigid demand for China's economic development. In 2013, the Chinese government launched the "Air Pollution Prevention and Control Action Plan" with $PM_{2.5}$ as the main control index, among which $PM_{10}$, $SO_2$, $NO_2$, $PM_{2.5}$, and CO pollutant emissions have been effectively improved. The increase in ozone concentration in ambient air has become more and more serious in recent years [1–3]. Ozone

numbers: 16BJY039), Harbin Normal University Graduate Student Innovation Project (grant numbers: HSDBSCX2023-12).

**Competing interests:** NO authors have competing interests.

plays a decisive role in the atmospheric environment, and high-altitude ozone is the Earth's protective film against ultraviolet burns on human skin. The environmental effects of surface ozone on humans are mainly due to the fact that ozone is very harmful to biological life and health [4], and high ozone concentrations lead to increased mortality from cardiovascular and respiratory diseases in the population [5, 6]. From the perspective of changes in urban ozone concentrations, the formation of ozone in the urban environment is complex. Ozone pollution caused by human activities is mainly caused by photochemical reactions of pollutants such as nitrogen oxides produced by production, construction, transportation, and fuel combustion [7]. At this stage, ozone pollution control is facing severe challenges. Since the "Twelfth Five-Year Plan" period, China has given great importance to the prevention and control of air pollution and has considered improving air quality as a major livelihood project. The concentration of a number of air pollutants focusing on $PM_{2.5}$ has dropped significantly, and the number of days of heavy air pollution has also decreased significantly [8]. However, near-surface Ozone pollution has become increasingly prominent [9], especially in summer has become the primary factor leading to excessive air quality in some cities [10]. Therefore, how to control the concentration of $PM_{2.5}$ and at the same time curb the increase of Ozone is the current air pollution prevention, and control of the urgent need to solve the problem, should attract the attention of the government and the public.

Urbanization is the process of economic, social, land, and population transformation from countryside to towns in the course of a country's development. Among them, the core of urbanization is population urbanization, and the carrier of urbanization is land urbanization [11]. With the intensification of atmospheric environmental pollution, scholars from all walks of life have studied the effect of pollution reduction and carbon reduction from multiple perspectives and found that urbanization is an important factor affecting the atmospheric environment [12]. "China's New Urbanization Report 2012" points out that the non-intensive and crude production of China's urbanization focuses on the quantity and scale of urban development, which ignores the value of environmental resources and is an important cause of the decline in the quality of the atmospheric environment. The problem of air pollution in China has changed from single pollutant pollution to compound pollution, which is increasing with the rapid development of urbanization in China [13, 14]. Civerolo et al. [15] used the SLEUTH model to extrapolate New York City's land cover data from 1990 to the next 2050 and predicted that future urbanization in New York will lead to elevated ozone concentrations. Scholars agree with the conclusion that urbanization has an impact on air pollution. Carbon emissions and haze are the main control gases. Scholars from all walks of life have done a lot of research on them, and the methods are diverse. From the existing literature, the STIRPAT model [16], Dubin model [17], DPSIR model [18], the geographically weighted regression model (GWR) [19], geographical detector [20], and so on are mainly used to study the development law of air pollution. The Theil coefficient, Theil coefficient, coefficient of variation, and spatial autocorrelation (global Moran's *I* index and local G coefficient) are often used to reveal the regional differences and spatial correlation of air pollution intensity. For example, Sun Yaohua et al. [21] explored the differences in carbon emission intensity between provinces and regions in China-based on the Theil index and found that there were differences in carbon emission intensity between provinces in China and accumulated. Zhao Yuntai et al. [22] divided the entire country into eight economic regions and used the Theil index, global autocorrelation Moran's *I*, and cold and hot spot analysis methods to explore the spatial evolution characteristics of regional carbon emission intensity. It was found that the widening difference between regions caused the widening difference of regional carbon emission intensity in the entire country, while the difference within the region was small. The research on the relationship between urbanization and air pollution mainly forms three conclusions: the positive

linear relationship between urbanization and air pollution [23, 24], the negative linear relationship between urbanization and air pollution [25], and the nonlinear curve relationship between urbanization and air pollution [26]. Among them, scholars have conducted empirical investigations on the different influencing factors of ozone on air pollution, but there are few studies on the impact of ozone distribution in air pollution from the perspective of urbanization. Chameides [27] believed that the urban heat island effect caused by urbanization is the main reason for the increase in urban ozone concentration. Many scholars use the WRF Chem model to check ozone pollution, and it shows that urbanization increases ozone concentration [28, 29]. Overall, the existing research results are relatively rich and provide an empirical basis for exploring the relationship between urbanization and air pollution. However, the exploration of urbanization on ozone pollution is still relatively small. In recent years, the causes of atmospheric pollution have strengthened from a purely environmental problem to a complex ecosystem problem of population, land, economy, and society. Current urbanization studies mostly use land cover data to select indicators and fail to consider other factors that influence ozone concentration. There is also limited research on the atmospheric environment effect from the perspective of urbanization. Therefore, understanding the characteristics of spatial and temporal changes in ozone and the driving mechanisms from the perspective of urbanization is essential to further promote the harmonious development of nature and society.

Considering all the above explanations, the present study proposes to address the following questions: i) The distribution characteristics of ozone in time and space. ii) Under the background of urbanization development, the driving mechanism of ozone pollution is analyzed (from the four urbanization development directions of economy, population, land, and society). iii) To unify the distribution of factors affecting ozone concentration by urbanization development in all provinces of the country. In addition, the distribution and driving mechanisms of ozone pollution from the perspective of China's urbanization are less frequently studied. To answer these questions above, we attempt to use ozone concentration data and statistical yearbook data of 31 provinces in China to study the driving mechanism of ozone pollution under the context of urbanization. We intensify on the impact of urbanization indicators on ozone and analyze the geographical spatial distribution of urbanization impact factors on the ozone impact index. By citing urbanization, hope to be more comprehensive on the causes of China's distribution characteristics of ozone pollution and how to solve this problem are discussed, and thus draw more accurate results and set forward more constructive suggestions. The intensification of atmospheric environmental pollution in China has attracted extensive attention from the government and academia, but the domestic existing literature, focuses on exploring the micro-channels of urbanization on ozone pollution, and the index selection is based on urban land cover data. The research on the impact of other influencing factors of urbanization on ozone concentration is indeed more insufficient. Therefore, this study aims to empirically analyze the impact mechanism and driving factors of ozone pollution in the context of urbanization in China using the dynamic spatial Durbin model and the spatio-temporal geographically weighted regression model. Producing a new explanation for the causes of the spatial and temporal distribution characteristics of ozone pollution, producing a basis for regional ozone and pollution prevention and control measures, has important theoretical and practical relevance for the government to formulate and improve atmospheric environmental protection policies, and how the government will carry out macro in the future.

## Materials and method

Scholars from all walks of life have conducted extensive research on the impact of urbanization on the atmospheric environment. Some scholars analyzed the influencing factors of

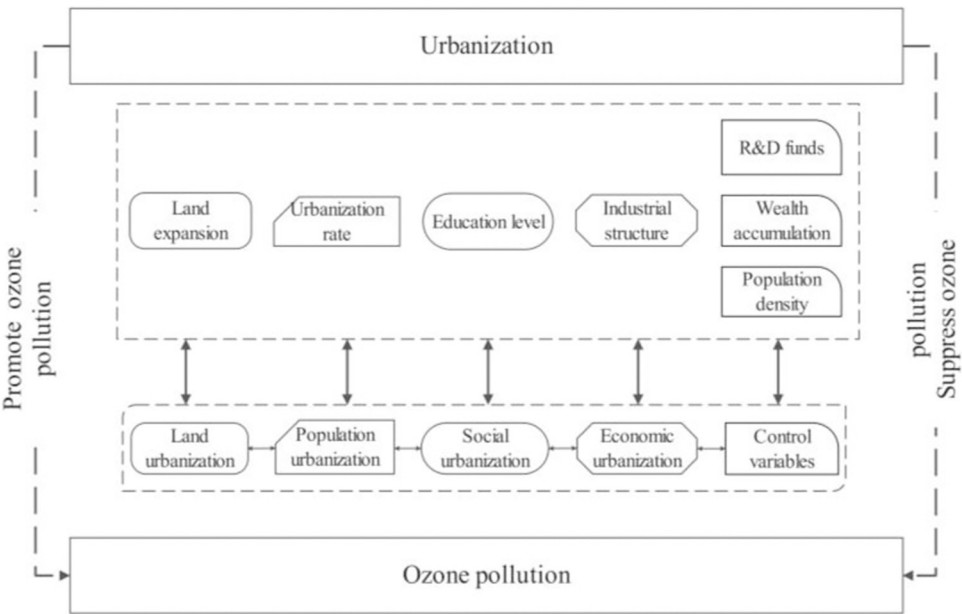

**Fig 1. Flow chart of urbanization ozone pollution.**

environmental pollutant emissions based on the STIRPAT model [30, 31]. Some scholars used the WRF Chem model to analyze ozone pollution by using the urban land cover rate as the research index in the process of urbanization [28, 29]. Some scholars have also analyzed the mechanism of urbanization on air pollution by dividing the urbanization process into economic, demographic, social, and land based on the OLS model [17]. Due to the complexity of today's air pollution, it is not sufficiently detailed to study the driving mechanisms of ozone pollution in terms of urban land cover or a single demo-graphically calculated urbanization rate the context of urbanization development in China. Based on the above method of dividing the urbanization process, combined with the actual situation in China, this paper breaks down the urbanization process into four parts, as shown in Fig 1, and analyzes the correlation mechanism between the two from the perspective of urbanization development in different situations.

## Methods

**Spatial correlation analysis.** This paper analyzes the spatial agglomeration effect of ozone based on global spatial autocorrelation. When $I < 0$, it indicates that ozone has a significant spatial negative correlation. When $I > 0$, it indicates that ozone has a significant spatial positive correlation, and the larger the value, the stronger the agglomeration effect of ozone. The basic formula is as follows:

$$I = [m\sum_{i=1}^{m}\sum_{j=1}^{m}\omega_{ij}(x_i - x)(x_j - x)]/[\sum_{i=1}^{m}\sum_{j=1}^{m}\omega_{ij}\sum_{i=1}^{m}(x_i - \bar{x})^2] \qquad (1)$$

Where, $m$, $x_i$ and $x_j$, $\omega_{ij}$, $x$ and $\bar{x}$ indicate the number of provinces in the country, the ozone concentration in the ambient air of $i$ and $j$ provinces, the spatial weight and the ozone concentration and its provincial mean, respectively. In this paper, three kinds of $31 \times 31$ order spatial weight matrices are constructed: Geographic distance weight matrix ($W_1$) whose element ($W_{ij}$) is expressed as the reciprocal of the distance between the provinces in region $i$ and the

provinces in region $j$. The economic distance weight matrix ($W_2$) whose element ($W_{ij}$) is expressed as the inverse of the absolute difference between the annual average per capita gdp in region $i$ and the annual average per capita gdp in region $j$. Geographic and economic distance nested matrix ($W_3$) whose element ($W_{ij}$) is expressed as the product of the reciprocal of the distance between the provincial capital of region $i$ and the provincial capital of region $j$ and the proportion of the annual average per capita gdp of region $i$ to the annual average per capita gdp of all the regions, with the expression $W_3 = (1-a)W_1 + aW_2$, and $a$ denotes the proportion of the economic weight, which takes the value of 0.5. $W_3$ denotes the space that takes into account the geographic and economic factors at the same time. Nested matrix.

The local Moran'$I$ index is able to compensate for local atypical features and can significantly express the clustering characteristics of ozone and its specific location in the spatial distribution. Its expression is:

$$I = [(x_i - x)/S^2] \times \sum_{j \neq i}^{n} \omega_{ij}(x_j - \bar{x})$$

$$S^2 = [\sum_{i=1} (x_i - \bar{x})^2]/m$$

(2)

Where, $x_i$ and $\bar{x}$, $m$, $\omega_{ij}$, $I$ indicate the attribute values of different provinces, the number of provinces, the spatial weight and the local Moran'$I$ value of provinces, respectively. At a certain level of significance, the local Moran'$I$ value index was classified into four spatial distribution patterns: high-low, low-high, high-high and low-low. The positive Moran'$I$ value index indicates low pollution and low agglomeration or high pollution and high agglomeration, and the negative value index indicates low pollution and high agglomeration or high pollution and low agglomeration.

**Ozone response effect model of urbanization process.** The IPAT model is mainly used to reflect the impact of human activities on atmospheric environmental pressure and to analyze the impact of ozone concentration ($I$), population size ($P$), affluence ($A$), and technological progress ($T$). We established STIRPAT model in the IPAT model, which is mainly used to study various environmental impact indicators at home and abroad. To study the relationship between the urbanization process and ozone concentration in ambient air in China, the traditional data model is improved [32–34] to test the relationship, direction, and intensity of mutual influence of spatial variables. Based on the STIRPAT model, a spatial econometric model of the relationship between the urbanization process and ozone concentration in ambient air is constructed [34]. The specific model expressions are as follows.

$$\ln I_{it} = \alpha_i + \beta_1 \ln P_{it} + \beta_2 \ln A_{it} + \beta_3 \ln T_{it} + \ln \varepsilon_{it}$$

(3)

In the formula, $\alpha$ represents the model coefficient. Where, $I$, $P$, $A$ and $T$ indicate he ozone concentration in the ambient air, population size, economic growth and technological level, respectively. $t$ represents the time, and $i$ represents the province. Where, $\beta_1$, $\beta_2$ and $\beta_3$ indicate the elasticity coefficients of population size, economic growth and technological level, respectively. $\varepsilon_{it}$ denotes the random disturbance term.

Based on the expression of the base model in Eq (3), combined with the definition of the factors influencing urbanization, the expression of the spatial measurement model is established as follows:

$$\ln I_{it} = \alpha_i + \beta_1 \ln pop_{it} + \beta_2 \ln gdp_{it} + \beta_3 \ln rd_{it} + \beta_4 \ln urd_{it} + \beta_5 \ln urd^2_{it}$$
$$+ \beta_6 \ln ind_{it} + \beta_7 \ln are_{it} + \beta_8 \ln stu_{it} + \ln \varepsilon_{it}$$

(4)

The main influencing factors include pop represents population density, *gdp* represents per

capita gdp, *rd* represents R&D intensity, *urd* represents urbanization rate, *urd²* represents the square of urbanization rate, *ind* represents industrial structure, *are* represents land expansion, *stu* represents education level. $\beta_1, \beta_2, \beta_3, \beta_4, \beta_5, \beta_6$ and $\beta_7$ represent their corresponding elasticity coefficients respectively. Where, $\alpha$, *t*, $\varepsilon_{it}$ and *I* represent the time, the random disturbance term, and *i* represents the province, respectively.

**Spatial-temporal geographically weighted regression model (GTWR).** The spatio-temporal geographically weighted regression model is extended on the basis of the geographically weighted regression model, which can effectively analyse spatial data heterogeneity. In order to deeply explore the spatial and temporal differences of the impact of urbanization on ozone in China, this paper analyzes the spatial distribution pattern of the influencing factors of ozone concentration in ambient air based on the spatio-temporal geographically weighted regression model. The specific formula is as follows:

$$Y_i = \beta_0(\mu_i, v_i, t_i) + \sum_{k=1}^{p} \beta_k(\mu_i, v_i, t_i)X_{ik} + \varepsilon_i; i = 1, 2, 3 \ldots n \qquad (5)$$

Where, $Y_i$, *n*, *k*, *t*, $\beta_0$, $X_{ik}$, $\beta_k$ $(\mu_i, v_i, t_i)$ *and* $\varepsilon_i$ indicate the ozone concentration value of the *i* province, the 31 provinces of China, the number of explanatory variables of the *i* province, the time, the time-space intercept term of the *i* province, the *k*th explanatory variable value of the *i* province, the regression coefficient of the *k*th explanatory variable of the *i* province, the error term. The selection of each variable and its measurement index in the formula is described as follows:

i) Industrial structure (*ind)* is an indicator to measure the progress of economic urbanization. ii) Population urbanization is the process of transformation from rural population to urban population. We use the urbanization rate (*urban*) calculated by demography as a variable to study. iii) The expansion of urban construction land area into rural areas mainly reflects the process of land urbanization. This paper measures the urban built-up area (*are*). iv) Social urbanization is mainly reflected in the process of urban culture infiltrating into rural areas and affecting rural lifestyles, which is measured by the ratio of college students (*stu*). In addition, based on the STIRPAT model, the following three control variables are selected: v) Population density (*pop*): The increase in population size will lead to energy consumption in cities. On the other hand, population agglomeration can improve the level of urban technology and improve the environment [35]. In order to analyze the actual impact of population size on ozone, we use population density (*pop*) as a variable to study. vi) Economic growth (*gdp*): Most scholars believe that there is a strong positive correlation between economic development and urbanization [17]. Economic growth can bring about changes in the structure of demand, thus driving the transfer of the primary industry to the secondary and tertiary industries, which depends on huge resource consumption. We use per capita gdp to characterize. The statistical description of the data (see Table 1) as follows.

## Data sources

Based on the inter-provincial panel data from 2015 to 2019, this paper takes 31 provinces and regions in China as research samples. According to the historical data of 268 prefecture-level cities released by China Air Quality Online Monitoring and Analysis Platform (https://www.aqistudy.cn/historydata/), the concentration data of $O_3$, $NO_2$, CO, $SO_2$, $PM_{10}$, $PM_{2.5}$ and AQI in each prefecture-level city were integrated at the provincial level, and the annual average value was used to calculate the concentration data of air pollutants in each province. $O_3$ (8h) is the ozone concentration data obtained by 8-hour ozone moving average concentration detection. We derived the data of urbanization variables and control variables from the National Bureau of Statistics (https://data.stats.gov.cn/easyquery.htm?cn=C01), and we derived all other relevant data from the statistical yearbooks of Chinese provinces.

**Table 1. Descriptive statistics (2015–2019).**

|  |  | Explanation | Mean | Std. Dev | Minimum | Maximum |
|---|---|---|---|---|---|---|
| $\ln O_3\_8h$ |  | 8-hour moving average ozone concentration ($\mu g/m^3$) | 4.4775 | 0.1289 | 4.1870 | 4.7765 |
| $\ln ind$ |  | The proportion of added value of the secondary industry (%) | -0.9362 | 0.2285 | -1.8228 | -0.6837 |
| $\ln gdp$ |  | The per capita GDP of the region (yuan / person) | 1.7190 | 0.4042 | 0.9618 | 2.7986 |
| $\ln urb$ |  | Proportion of urban population in total population (%) | -0.5511 | 0.2086 | -1.2809 | -0.1244 |
| $\ln pop$ |  | Population density (persons / km2) | 7.8842 | 0.3872 | 7.0353 | 8.6152 |
| $\ln rd$ |  | R&D expenditure (ten thousand yuan) | 14.2589 | 1.7212 | 7.864 | 16.9574 |
| $\ln are$ |  | Built-up area (square kilometers) | 7.2066 | 0.8149 | 4.8283 | 8.6264 |
| $\ln stu$ |  | College student ratio (%) | -4.4372 | 0.3035 | -5.1126 | -3.7097 |

## Results

### Characterization of ozone spatio-temporal evolution

**Characterization of ozone time evolution.** Comparison of air quality index AQI and characteristics of $PM_{2.5}$, $PM_{10}$, $SO_2$, CO, NO, and $O_3$ concentration changes in ambient air in China from 2015 to 2019 (Fig 2). From the perspective of concentration change trend, ozone concentration continued to rise in 2015–2017, decreased slightly in 2017–2018, and then continued to rise again. The lowest emission concentrations of AQI and $PM_{2.5}$, $PM_{10}$ and $NO_2$ were in 2018, slightly increased in 2019, and the highest emission concentration was in 2015. The emissions of AQI, $PM_{2.5}$, $PM_{10}$, $SO_2$, CO and $NO_2$ pollutants decreased by $8.85\mu g/m^3$, $13.99\mu g/m^3$, $22.47\mu g/m^3$, $13.22\mu g/m^3$, $0.29mg/m^3$ and $1.78\mu g/m^3$ in 2019 compared with 2015. The overall air quality of the city has improved. Only ozone concentration was elevated by $8.39\mu g/m^3$ in 2019 compared to 2015, which is a severe upward trend compared to other air

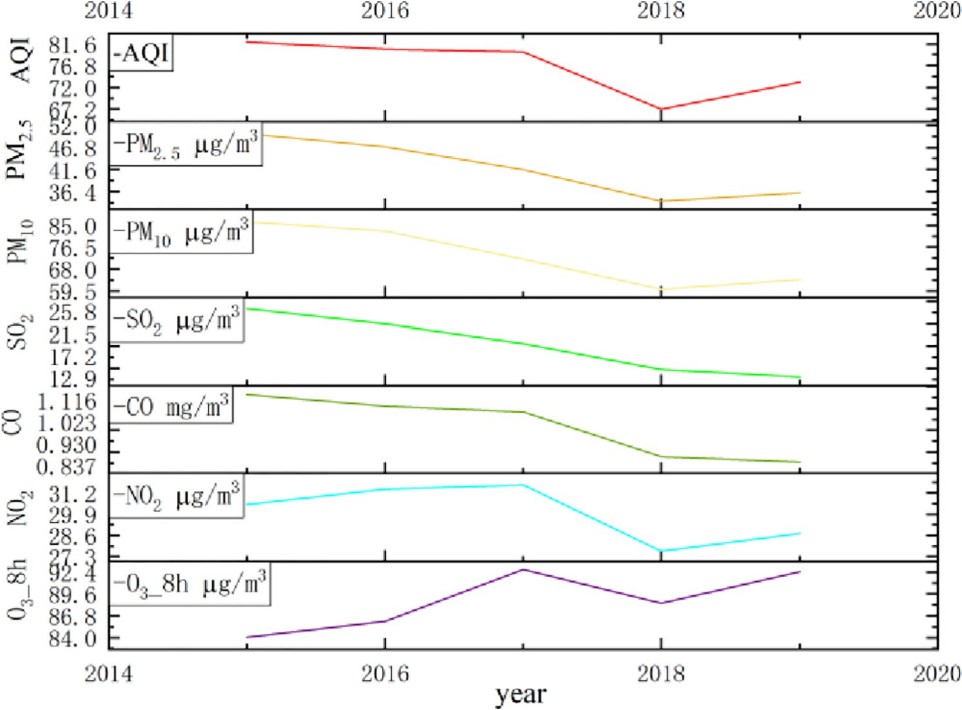

**Fig 2. 2015–2019 AQI and changes in the concentration of various pollutants in ambient air.**

**Table 2. 2015–2019 national ozone pollution level Moran'*I* index and its inspection results.**

| Moran'*I* | 2015 | | 2016 | | 2017 | | 2018 | | 2019 | |
|---|---|---|---|---|---|---|---|---|---|---|
| | *I* | *P* | *I* | *P* | *I* | *P* | *I* | *P* | *I* | *P* |
| ln$O_3$_8h | 0.027 | 0.220 | 0.128 | 0.019 | 0.209 | 0.001 | 0.166 | 0.005 | 0.244 | 0.000 |
| ln*gdp* | 0.037 | 0.180 | 0.028 | 0.213 | 0.025 | 0.222 | 0.018 | 0.251 | 0.014 | 0.269 |
| ln*ind* | -0.039 | 0.465 | -0.042 | 0.451 | 0.063 | 0.339 | -0.050 | 0.403 | -0.057 | 0.366 |
| ln*pop* | -0.081 | 0.269 | -0.128 | 0.112 | 0.146 | 0.073 | -0.153 | 0.060 | -0.121 | 0.128 |
| ln*urb* | 0.026 | 0.212 | 0.027 | 0.209 | 0.030 | 0.198 | 0.030 | 0.192 | 0.035 | 0.175 |
| ln*urb*$^2$ | 0.006 | 0.271 | 0.007 | 0.267 | 0.010 | 0.253 | 0.011 | 0.240 | 0.015 | 0.211 |
| ln*are* | 0.080 | 0.066 | 0.076 | 0.074 | 0.064 | 0.097 | 0.061 | 0.104 | 0.058 | 0.113 |
| ln*stu* | 0.335 | 0.000 | 0.355 | 0.000 | 0.366 | 0.000 | 0.374 | 0.000 | 0.374 | 0.000 |
| ln*rd* | 0.222 | 0.218 | 0.029 | 0.196 | 0.036 | 0.169 | 0.034 | 0.182 | 0.030 | 0.194 |

pollutants. This shows that ozone pollution researches are necessary for formulating atmospheric environmental policies.

**Characterization of ozone spatial evolution.** The presence or absence of spatial autocorrelation features between variables determines whether the STARPAT model of Eq (4) can incorporate spatial effects. In order to accurately analyze the impact of urbanization on ozone pollution and test whether there is spatial autocorrelation between urbanization rate and ozone pollution, we calculated the Moran'*I* values for both as shown in Table 2. We can observe from the table that environmental pollution levels in 2015–2019 mostly have positive Moran'*I* values. This indicates that there is a significant positive spatial autocorrelation in the level of ozone pollution among 31 provinces in China. In addition, the Moran'*I* value of ozone in 2015 was 0.027, which rose to 0.244 in 2019, indicating that the spatial agglomeration effect of ozone pollution in China was becoming more and more obvious.

The results of Fig 3 further show that the distribution characteristics of near-surface ozone in China are "east-heavy, west-light," "short in the south and high in the north". The spatial distribution trend of diffusion to central China and coastal cities is more prominent in the Beijing, Tianjin, Hebei, and Shandong provinces. Significant spatial heterogeneity characterizes the inter-province. Hebei Province, Tianjin City, and Shandong Province are the key areas of

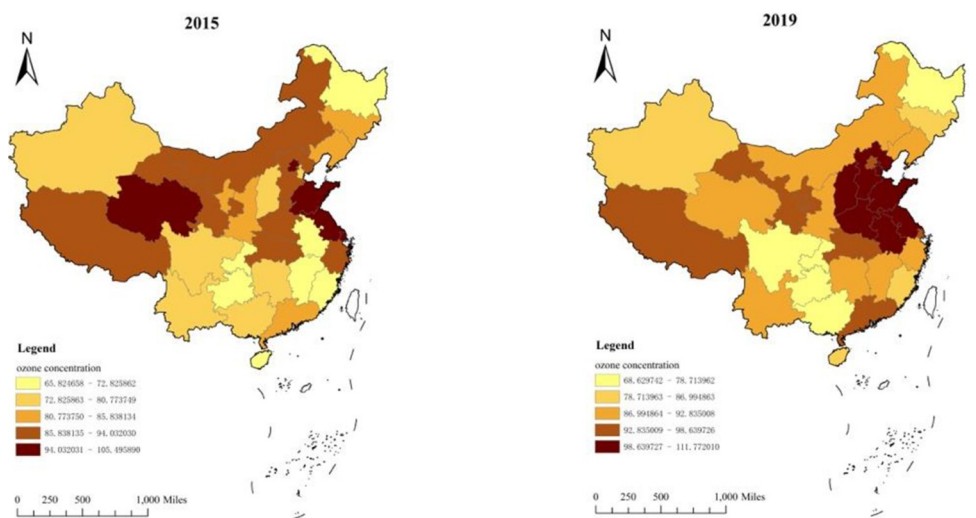

**Fig 3. The spatial distribution of ozone concentration in China's provinces in 2015 and 2019.**

the petrochemical and organic chemical industry in China, and the stock of coal-fired boilers is relatively large. The region has dense traffic, developed logistics, and huge emissions of diesel trucks and fuel vehicles. These factors have expanded VOCs and nitrogen oxide emissions in these areas, forming a medium and long-term pressure on ozone prevention and control in the region. In 2015, the high-value areas of ozone concentration in China were distributed across the eastern region, mainly in Beijing, Shanghai, Shandong, Jiangsu, Qinghai, Zhejiang, Gansu, and Inner Mongolia. In Shanghai, the average annual concentration reached $105\mu g/m^3$. By 2019, the concentration of ozone in China increased compared with 2015, and ozone pollution expanded rapidly in the east, mainly concentrated in North China and East China. Among them, Hebei Province, Tianjin City, and Shanxi Province in North China are the high-value areas of ozone pollution concentration growth in China. The near-surface ozone concentration in Tianjin increased from $77 \ \mu g/m^3$ in 2015 to $106 \ \mu g/m^3$ in 2019. This is because more developed industries, dense populations, and higher emissions of ozone precursors from motor vehicles and industrial sources typically characterize these areas [5].

## Response of ozone pollution to urbanization

**Response results.** In order to determine the specific form of the model, this paper carried out the LM test, LR test, and Hausman test. According to the LM results in Table 3, LM rejects the null hypothesis, and the results tend to be the spatial error model (SEM). Further, the LR test is used to verify the model, and it is found that the null hypothesis is rejected at the 1% level. The Hausman test showed that the null hypothesis was rejected at the 1% level. Therefore, the fixed effect spatial error model (SEM) was finally selected for the experiment.

Based on the SEM partial differential method, the spillover effect model of urbanization on ozone pollution is constructed (Table 4). The total effect can be decomposed into two parts. One is the direct effect, which indicates the impact of local urbanization development on ozone pollution in the region. The other is the indirect effect, which indicates the impact of local urbanization development on ozone pollution in neighboring areas. Overall the direction of the coefficients of the indicators of the total effect is consistent with the indirect effect. The results of the analysis of the specific individual indicators are:

i) The first-order coefficient of the total effect of urbanization rate (*urb*) is 29.09, and the second-order coefficient of urbanization rate is 17.17. All have significant spillover effects at the 5% level. Production, transportation, fuel combustion and other activities caused by the scale effect of the urban population are the main causes of ozone pollution in the region [12].

ii) The coefficient of economic growth (*gdp*) is positive, with a coefficient of 0.08 for the direct effect and 0.84 for the indirect effect, indicating that economic development contributes more to ozone pollution in neighboring areas than locally. This may be due to the pressure of

**Table 3. LM test of spatial panel model.**

| LM test | | | |
|---|---|---|---|
| | **Statistic** | **df** | **_p_-value** |
| Spatial error | | | |
| Moran's *I* | 113.175 | 1 | 0.000 |
| Lagrange multiplier | 8.510 | 1 | 0.004 |
| Robuet Lagrange multiplier | 8.680 | 1 | 0.003 |
| Spatial lag | | | |
| Lagrange multiplier | 0.826 | 1 | 0.364 |
| Robuet Lagrange multiplier | 0.995 | 1 | 0.319 |

**Table 4. Spatial error model under nested matrix.**

| Variable | Direct effect | | Indirect effect | | Gross effect | |
|---|---|---|---|---|---|---|
| | coefficient | t | coefficient | t | coefficient | t |
| ln*gdp* | 0.082 | 1.14 | 0.84 | 1.28 | 0.92 | 1.33 |
| ln*ind* | 0.24** | 2.56 | 1.31*** | 3.16 | 1.54*** | 3.56 |
| ln*pop* | -0.04 | -0.98 | 0.94* | 1.68 | 0.91 | 1.56 |
| ln*urb* | 0.10 | 1.31 | 28.09** | 2.30 | 29.09** | 2.31 |
| ln*urb²* | 0.83* | 1.93 | 16.34** | 1.99 | 17.17** | 2.03 |
| ln*are* | 0.03 | 0.52 | 0.02 | -0.05 | 0.06 | 0.12 |
| ln*stu* | -0.10 | -0.65 | -6.11** | -2.33 | -6.22** | -2.31 |
| ln*rd* | 0.04 | 1.08 | -0.13 | -0.30 | -0.08 | -0.19 |

Note:* $p < 0.1$

** $p < 0.05$

*** $p < 0.01$.

regional emission reduction, which brings pollution problems to neighboring areas in the form of pollution industry transfer, resulting in higher ozone pollution concentration in neighboring areas than in local areas.

iii) Industrial structure (*ind*) is positively correlated with ozone concentration and is significant at 1 per cent level. It shows that for every 1% increase in industrial structure the ozone concentration in the area increases by 1.54$\mu g/m^3$. China's coast is a key agglomeration of heavy industrial industries. In the process of industrial development of the ecological environment is poor, the industrial structure of the heavy or transport structure and other factors are important reasons for the increase in regional ozone concentrations [36]. Emission reduction policies tailored to different regional conditions are important for the ozone environment [37].

iv) The direct effect coefficient of population density (*pop*) is -0.04, the indirect effect coefficient is 0.94, and the ozone pollution in the adjacent areas is significant at the 10% level. It shows that the agglomeration of population in the region can reduce the local ozone pollution, but it can promote the ozone pollution in the adjacent area. The reason is that population agglomeration can promote the improvement of technological level and reduce ozone concentration to a certain extent, but the activities of motor vehicle exhaust and industrial emissions caused by the scale effect of population will lead to the increase of ozone concentration in ambient air. In addition, the influx of population to economically developed areas and the neighboring underdeveloped areas due to population loss leads to lower emission reduction technology than developed areas, which may also lead to increased ozone pollution.

v) Some scholars believe that urban land expansion aggravates ozone pollution. The coefficients of land expansion (*are*) in this study are positive, which is consistent with most research conclusions [17, 27, 28]. It shows that the expansion of an urban scale brings huge energy consumption and has a positive impact on the increase of ozone concentration. Factors such as unreasonable urban construction and management measures can lead to a decline in urban environmental quality.

vi) The coefficients on educational attainment (*stu*) were all negative, and the total effect coefficient was -6.22, significant at the 5 per cent level, indicating that a 5 percent point increase in the proportion of university students was associated with a reduction in ambient air ozone concentrations of 6.22$\mu g/m^3$. This shows that the ratio of college students represents the level of social urbanization in China, which affects social consciousness to a certain extent [17] and inhibits ozone pollution.

**Table 5. Change trend of regression coefficient of factors affecting ozone concentration.**

| Variable | Maximum | Minimum | Median | 25% percentile | 75% percentile |
|---|---|---|---|---|---|
| *are* | 0.013 | -0.013 | 0.004 | 0.000 | 0.008 |
| *pop* | 0.003 | -0.005 | 0.001 | 0.000 | 0.001 |
| *urb* | 62.540 | -123.205 | 9.674 | -37.422 | 24.124 |
| *ind* | 96.299 | -78.048 | 36.064 | 7.772 | 58.202 |
| *gdp* | 7.900 | -4.660 | 1.760 | 1.250 | 3.910 |
| *rd* | 4.1763E-06 | -0.000002835 | -9.437E-07 | -1.63405E-06 | -1.442E-07 |
| *stu* | 1066.860 | -694.910 | -180.780 | -352.260 | 9.830 |
| Bandwidth | | | | 0.206 | |
| AICc | | | | 1168.130 | |
| $R^2$ | | | | 0.595 | |
| $R^2$Adjusted | | | | 0.575 | |
| Trace_of_SMatrix | | | | 40.753 | |

vii) The effect of R&D intensity (*rd*) on local ozone concentrations is positive and the effect on neighboring areas is negative. This may be the reason technological advances are based on the actual production process [38]. In terms of technology, technology can be divided into abatement technology and production technology, and the direction of research and development funding greatly affects ozone concentrations.

**Regression results.** The impact of urbanization development on the increase of ozone concentration in ambient air in China has spatial correlation and heterogeneity. So, ArcGIS software was used to construct a regression model. Before the GTWR model, it is necessary to test the collinearity of each index. According to the test results, the mean value of VIF is 4.14, the maximum value of VIF is R&D intensity (*rd*) of 8.22, and all the indicators pass the covariance test.

Table 5 shows that some of the variables are contributing to a significant increase in ozone in ambient air. The range of the upper and lower quartiles of the index that promotes the increase of ozone concentration in ambient air is *are* (0–0.008), *pop* (0–0.001), *ind* (7.772–58.202) and *gdp* (1.25–3.91). The inhibitory effect of *rd* and *stu* on ozone decreases progressively, with the upper and lower quartiles of education ranging from -352.26 to 9.83, and the influence direction of the lower quartile became positive. The inhibitory effect of *stu* and *ind* on ozone concentration began to decrease. On the contrary, the promotion effect of *urb*, *are* and *gdp* on ozone concentration was gradually increasing.

There are spatial differences in the influence of different factors on ozone concentration (Fig 4) The importance of the influencing factors of ozone concentration from large to small is *stu*, *ind*, *urb*, *gdp*, *are*, *pop* and *rd*. Specifically:

i) The high-value areas of the influence of *stu* on ozone are mainly distributed in Tibet and Xinjiang. The ozone concentration in Xinjiang and Tibet does not exceed the national standard, but according to the National Bureau of Statistics, the education level in Xinjiang and Tibet is low, and the level of environmental supervision is always backward [39], and the backward education system restricts the rational development of urban environment.

ii) There is a positive correlation between *ind* and ozone. In terms of spatial distribution, it decreases inward with the north and south as the boundary. The high-value areas are mainly distributed in Inner Mongolia, the three northeastern provinces (Heilongjiang, Jilin, Liaoning) and Tibet, and the average concentration rate of energy-intensive industries in the high-value areas is always on the rise [40]. It can be seen that the unbalanced development of China's industrial structure is an important factor contributing to the aggravation of ozone pollution.

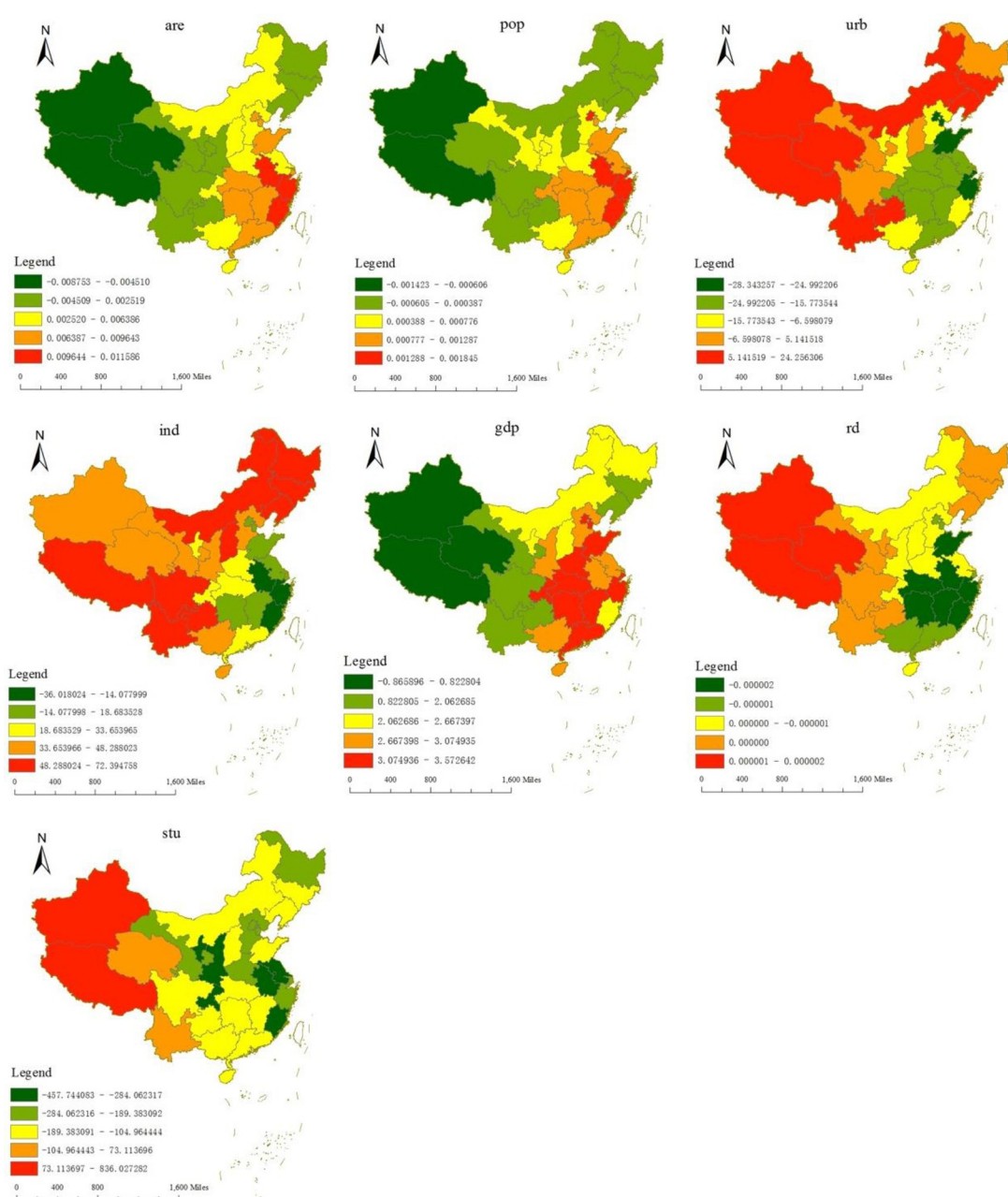

**Fig 4. Distribution of regression coefficients of factors affecting ozone from 2015 to 2019.**

iii) The sensitive areas of the impact of *urb* on ozone concentration are mainly distributed in the northern region, and the degree of influence on different provinces is obviously different. The distribution pattern is mainly decreasing from the northern region to the southern region. Specifically, it is concentrated in Inner Mongolia, Xinjiang, Tibet, Qinghai, Jilin, Liaoning, Yunnan and Guizhou.

iv) The high-value areas of the impact of economic growth on ozone concentration are mainly distributed in Beijing, Guangdong, Hunan, Hubei, Zhejiang, and other places. The average values of the coefficients are 3.46, 3.42, 3.51, 3.33 and 3.34, respectively, showing a spatial pattern of decreasing outwards with Beijing and Guangdong as the boundary. Economic

development is an important factor leading to atmospheric environmental pollution, and high-value areas are mainly distributed in cities with rapid economic development [41].

v) The high-value areas of the impact of *are* on ozone concentration mainly include Zhejiang, Fujian, Anhui, and Shanghai, with an average coefficient of 0.0114, 0.0112, 0.0116, and 0.009, respectively. Overall there is a spatial pattern of decreasing gradient in all directions with Zhejiang, Fujian and, Anhui as the centre. The reason is that the development of land urbanization leads to the change land use type, and the urban thermal pollution caused by the increasing impervious surface will lead to the aggravation of ozone pollution [42, 43].

vi) The high-value areas of the impact of *pop* on ozone concentration are mainly distributed in Beijing, Zhejiang, Anhui, and Fujian, mainly decreasing outward from the southern region. The change rates of *pop* from 2015 to 2019 are -26%, -3.6%, -1.2% and 9%, respectively. Except for the large change in the rate of change of *pop* in Beijing, the changes were minimal, and only in Fujian did the *pop* increase, and the population effect did not play a significant role in the increase of ozone concentration.

vii) The level of technological development is the determinant of human impact on climate change [44]. From this study, the impact of *rd* on ozone concentration is low, and the sensitive areas are mainly distributed in Xinjiang, Tibet, and Qinghai. According to the data of the National Bureau of Statistics, Xinjiang, Tibet, and Qinghai are too low in R&D investment compared with other provinces, and low-level emission reduction technologies are not used to solve the contradiction between urbanization development and ozone pollution.

## Discussion

### Discussion on urbanization and ozone

Urbanization refers to the country in the development process of economic, social, land and population from rural to urban transformation process. Among them, the core of urbanization is population urbanization, and the carrier of urbanization is land urbanization. China's urban development has entered a new normal, and over time, it will add more series to the ranks of rural-urban transformation. Although the rate of urbanization in China has been gradually slowing down in recent years, the fact remains that urban factors are gradually penetrating the countryside, and the countryside is gradually transforming into old urban areas. Urban cultural development is gradually affecting the countryside by moving from contact transmission to stimulation transmission. The transformation of rural areas into urbanization is most notably reflected in the economy, which may lead to a decline in the share of the agricultural industry in the transformation process of China's urbanization development. In addition, urbanization has, to a certain extent, led to an increase in the rural exodus, with a large part of the population choosing to migrate from low-income to high-income areas, thus leading to an increase in the contraction of the rural population. The conclusion of increasing ozone concentration in China is consistent with the conclusion of Han et al. [14]. In recent years, the cause of air pollution has developed from a simple environmental problem to a complex ecosystem problem of population, land, economy, and society. In previous studies, urban land cover data mostly verified the selection of urbanization indicators, ignoring the impact of other influencing factors of urbanization on ozone concentration. Therefore, this paper analyzes the driving mechanism of ozone pollution from a new urbanization perspective in the context of rising national ozone concentration, to provide a reference for China's atmospheric environmental protection policy.

### Discussing the impact of urbanization on ozone

The influencing factors of ozone are discussed from four aspects of urbanization development. The results show that the impact of urbanization development on ozone has a positive effect

on the whole. Land expansion and population size effect have a positive effect on the aggravation of regional ozone pollution, which is consistent with the research results of most researchers [12, 20, 27, 28, 45]. Economic development is one of the key influences driving the urbanization of the region and deserves focused consideration. China's economic development is gradually changing from high energy consumption and low utilization rate to low energy consumption and high utilization rate. The transformation of industrial structure has important statistical significance in the regression model.

Social urbanization shows that the education level of a region can reflect the social consciousness of a region to a large extent. Highly educated areas tend to have more knowledge about environmental protection and contribute more to environmental pollution. For example, according to the conclusion of the geographically weighted regression model, the high-value area of the regression coefficient of education level is mainly located in Xinjiang, while the education level in Xinjiang and Tibet is relatively backward, which can indirectly explain the important position of education level in environmental pollution. From the technical level, technology can be divided into emission reduction technology and production technology. The direction of R&D investment affects ozone concentration to a large extent. Surprisingly, increasing R&D intensity has a positive effect on local ozone concentration, but a negative effect on neighboring areas. It has been proven through research that the majority of R&D funds are used in production technology. Therefore, it is not desirable to pursue urbanization to achieve rapid economic development in China. The primacy of the natural environment is fundamental and basic to the development of modern China. This study suggests that improving education, investing in emission reduction technologies, and promoting industrial transformation are effective in lowering ozone concentrations.

## Discussion on control variables and research methods

Li et al. [31] used the data of 31 provinces from 2000 to 2017 as samples to test the impact mechanism of urbanization on environmental pollution. The results show that urbanization has a significant spatial spillover effect on environmental pollution, and the spillover effect between regions is greater than the local effect within the region. Shao et al. [38] tested the heterogeneous impact of the two urbanization promotion modes of compact intensive and scale expansion on haze pollution. Lou et al. [46] conducted an empirical study in Wuyi County. The results showed that there was a significant positive correlation between the urbanization development index (except for total industrial output value) and ozone mass concentration, and urbanization development aggravated ozone pollution. All the above studies illustrate that urbanization due to the economy and land leads to increased ozone pollution. However, this paper argues that it is unrealistic to reflect the impact of urbanization on ozone in terms of economics and land alone. In recent years, the causes of atmospheric pollution have strengthened from a purely environmental problem to a complex ecosystem problem of population, land, economy, and society. The urbanization indicators selected in this study are more diversified, not only validated with urban land cover data but also considering the impact of other influencing factors on ozone pollution concentration. Therefore, it differs from previous studies. In this work, the author not only analyzes the changes of various air indicators over the years from the perspective of time, but also uses spatial econometric models and geographically weighted regression models to analyze the impact of urbanization indicators on ozone, and analyzes the impact of urbanization impact factors on ozone Impact index on the geographical spatial distribution. The results of urbanization Moran's I show that the autocorrelation between ozone and various influencing factors (land, population, economic, and social urbanization) is statistically significant. This makes the study more comprehensive.

In terms of research methodology, in order to discern the specific form of the model, this study conducted LM test, LR test with Hausman test. The LM rejected the original hypothesis, and the results favored the spatial error model (SEM), and the model was further validated by the LR test, which was found to reject the original hypothesis at the 1% level. And the original hypothesis is rejected at 1% level as shown by Hausman test. The fixed-effects spatial error model (SEM) was finally chosen for the experiment. The spatial error model (SEM) is based on selected criteria (log-likelihood, AIC, SBC) and is relatively widely used because it incorporates both endogenous and exogenous interaction effects.

## Conclusion

After analyzing the distribution and spatial and temporal characteristics of ozone pollution in China, this paper explores the driving mechanism of ozone pollution in the context of urbanization. Using the spatial econometric model and quantitative method, this paper analyzes the regional differences, temporal and spatial distribution patterns, influencing factors, and spatial spillover effects of ozone pollution in China, with the conclusions as follows:

First, the lowest value of ozone concentration in China's ambient air from 2015 to 2019 was in 2015, and the average annual increase was $1.68\mu g/m^3$. There is a significant spatial autocorrelation in the level of ozone pollution. Among them, Beijing, Shanghai, Tianjin, Shandong, Henan, Hebei, Inner Mongolia, Anhui, Liaoning and Jiangsu are always the high value areas of ozone concentration, and the spatial distribution of the agglomeration area in the eastern expansion of the "east-heavy and west-light" and "south-short and north-high" characteristics, which indicates that ozone pollution has obvious regional and agglomeration characteristics.

Second, selecting a spatial econometric model to quantitatively identify the key factors influencing ozone concentration in ambient air, we found that *stu* and *rd* have a negative effect on the increase of ozone concentration, while *ind*, *urb*, *gdp* and *are* have a positive effect on the increase of ozone concentration. Studies have shown that there are not only direct effects that affect the level of local ozone pollution, but also indirect effects that affect the level of adjacent ozone pollution also known as spillover effects.

Last, there is also spatial heterogeneity in the factors influencing the increase in ozone concentration. The regression model analysis shows that the inhibitory effect of *stu* and *rd* on the increase of ozone concentration has been reduced, and the promotion effect of *urb* rate, *are* and *gdp* on the increase of ozone concentration has accumulated. The importance influencing factors of ozone concentration from large to small are *stu*, *ind*, *urb*, *gdp*, *are*, *pop* and *rd*.

Based on the research conclusions of this paper, to further reduce ozone pollution, it is necessary to reduce the negative effects brought about by the development of urbanization in China, so as to promote the high-quality development of urban economy. Therefore, we propose the following countermeasures and suggestions:

First, it is necessary to strengthen and improve the system design of ozone pollution control. Ozone pollution sources are mainly from nitrogen oxides and VOC precursors. It should include nitrogen oxides and VOC precursors in the national emission reduction index system to control their emissions. Since the precursors responsible for ozone pollution also contribute to $PM_{2.5}$ emissions, combining ozone pollution control and $PM_{2.5}$ control can enhance interactive data monitoring between the two and facilitate data sharing. Ozone pollution is optical pollution, and the amount of pollution is large when the temperature is high. Therefore, in the process of controlling climate warming, it is also necessary to consider the influence and change of ozone concentration and strengthen the synchronization of climate change control and ozone control.

Second, establish a regional ozone pollution collaborative governance mechanism. Ozone pollution has significant spatial dependence characteristics. The development strategy of

environmentally friendly urbanization formulated separately in any region has a limited inhibitory effect on ozone pollution. Only by strengthening inter-regional coordination and cooperation and implementing joint prevention and control can ozone be treated more effectively. The design and improvement of inter-regional ecological protection compensation mechanisms should be strengthened, and financial transfers to areas with serious ozone pollution should be increased.

Third, develop differentiated industrial structure adjustment strategies. There are huge differences in regional economic development level, factor endowment, industrial structure, and urbanization degree, and ozone pollution shows obvious spatial heterogeneity. Therefore, we should adopt differentiated industrial restructuring strategies to achieve the reduction of ozone pollution. For areas with high urbanization levels, we should accelerate the elimination of high pollution and high energy consumption industries, promote the transformation of resource-intensive industries and labor-intensive industries to technology-intensive industries, vigorously develop high-tech industries and modern service industries such as finance and logistics, and accelerate the proportion of the tertiary industry in the national economy.

Last, strengthen the research development and promotion of environmental protection technology. Technological progress is an important means to improve the efficiency of resource utilization and improve the ecological environment. In the process of urbanization, it is necessary to accelerate the research and development, application and promotion of environmental protection technology, and realize the coordinated development of urbanization and ozone governance.

## Supporting information

**S1 Table. Ozone concentration and urbanization level in China, 2015–2019.**
(XLS)

**S1 Data.**
(XLSX)

## Author Contributions

**Conceptualization:** Xiang-Li Wu.

**Data curation:** Li-Min Wang, Zi-Yi Ran, Heng-Yu Wang, Li-Bin Zhao.

**Formal analysis:** Li-Min Wang.

**Funding acquisition:** Li-Min Wang, Xiang-Li Wu.

**Investigation:** Zi-Yi Ran.

**Methodology:** Li-Min Wang, Zi-Yi Ran.

**Resources:** Xiang-Li Wu.

**Software:** Li-Min Wang.

**Supervision:** Xiang-Li Wu.

**Validation:** Heng-Yu Wang.

**Writing – original draft:** Li-Min Wang.

**Writing – review & editing:** Li-Min Wang, Zi-Yi Ran.

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
