## [Decision Letter · Decision Letter 0]

19 Nov 2023

PONE-D-23-35953Spatialtemporal evolution characteristics of ozone in China and its response to urbanizationPLOS ONE

Dear Dr. Wu,

Thank you for submitting your manuscript to PLOS ONE. After careful consideration, we feel that it has merit but does not fully meet PLOS ONE’s publication criteria as it currently stands. Therefore, we invite you to submit a revised version of the manuscript that addresses the points raised during the review process.

We look forward to receiving your revised manuscript.

Kind regards,

Shazia Rehman, Ph.D.

Academic Editor

PLOS ONE

Journal Requirements:

   "This work was supported by the National Social Science Foundation of China (grant numbers: 16BJY039), Harbin Normal University Graduate Student Innovation Project (grant numbers: HSDBSCX2023-12)."

3. We note that Figure 4 in your submission contain map/satellite images which may be copyrighted. All PLOS content is published under the Creative Commons Attribution License (CC BY 4.0), which means that the manuscript, images, and Supporting Information files will be freely available online, and any third party is permitted to access, download, copy, distribute, and use these materials in any way, even commercially, with proper attribution. For these reasons, we cannot publish previously copyrighted maps or satellite images created using proprietary data, such as Google software (Google Maps, Street View, and Earth). For more information, see our copyright guidelines: http://journals.plos.org/plosone/s/licenses-and-copyright.

a. You may seek permission from the original copyright holder of Figure 4 to publish the content specifically under the CC BY 4.0 license.  

Reviewers' comments:

Reviewer's Responses to Questions

**Comments to the Author**

1. Is the manuscript technically sound, and do the data support the conclusions?

Reviewer #1: Yes

Reviewer #2: Yes

2. Has the statistical analysis been performed appropriately and rigorously? 

Reviewer #1: Yes

Reviewer #2: Yes

3. Have the authors made all data underlying the findings in their manuscript fully available?

Reviewer #1: Yes

Reviewer #2: Yes

4. Is the manuscript presented in an intelligible fashion and written in standard English?

Reviewer #1: Yes

Reviewer #2: Yes

5. Review Comments to the Author

Reviewer #1: As urbanization in China continues, ozone pollution is becoming increasingly severe and has become one of the key challenges in atmospheric environmental management. This paper analyzes the driving factors behind China's ozone pollution from the perspective of urbanization, which shows certain innovation in topic selection. The research framework is reasonable with rich data support, and the conclusions can provide references for regional atmospheric environmental protection policies. Overall, this is an important and practically valuable study, and the structure and content of the paper are well organized. Therefore, I would recommend some minor revisions:

(1) The introduction needs to be improved, and it is necessary to expose the innovations that this study brings compared with other urbanization or air pollution studies. We also encourage the authors to add some references to improve the introduction according to the research gaps and the needs of the study.

[1] Cao N, Luo N, Peng Y, et al. Evaluation of Urbanization Development Dynamics Based on Quadrilateral Diamond Structure Model: Evidence from Chongqing, China, in the New Era[J]. Journal of urban planning and development, 2023,149(3). https://doi.org/10.1061/JUPDDM.UPENG-4413

[2] Sun P, Luo N. Comparative Analysis on the Quality of Urbanization Structure of Central Cities in Southwest Economic Core Area of China: Taking Chengdu-Chongqing as Examples[J]. Scientia Geographica Sinica, 2021,41(6): 1029-1039. DOI:10.13249/j.cnki.sgs.2021.06.011

(2) Please ensure the accuracy of data sources: the author uses air pollution data at the prefecture-level city, so the process of aggregating such data to the provincial level requires further explanation. Also, the URL in line 197 contains errors, please check and correct it.

(3) "Fig 1. Flow chart of urbanization ozone pollution" needs correction. The relationships and connections between different components in Fig. 1 are not clearly characterized. I suggest the author further refine the contents in this figure, especially by establishing clearer links between aspects of urbanization and specific variables (e.g. industrial structure, urbanization rate). This could better explain the rationale behind selecting the independent variables, and illustrate the mechanism linking urbanization and ozone pollution.

(4) The "Characterization of ozone spatio-temporal evolution" in the Empirical Results section does not combine time and space in the analysis, making the contents and results less clear. I suggest the author reorganize this part, for example, combining Table 3 and Fig. 3 to present the results on a map, which could better demonstrate the characteristics of different clustering types.

(5) In the Response Results section, please add validation results of the spatial econometric models (in a table form); also, only very few indicators in Table 4 pass the significance test, and the Coefficients and t-values are inconsistent in signs, please double check and correct these issues.

(6) Suggest adding policy implications in the Discussion or Conclusion section to highlight the practical significance of this study, and echo the problem the author aims to address - "to provide a reference for the government to formulate and improve atmospheric environmental protection policies".

(7) There are issues regarding formatting conventions (e.g. characters in formulas should be italic, superscripts for units, the "3" in O3 should be a subscript, etc.), table/figure formatting (e.g. font size in figures is too small, affecting readability), and other details that could be standardized. I will not list them one by one here, but the author should thoroughly proofread and edit the manuscript.

Reviewer #2: This paper is interesting and find some valuable conclusion. However, this paper has to make the following revisions.

(1) In the "Introduction" section, the introduction expounds too much very general and well-known information, and the importance and significance of the research are not highlighted.

(2) In the "Introduction" section, The author lacks references when expounding the research background.

(3) For readers to quickly catch your contribution, it would be better to highlight major difficulties and challenges, and your original achievements to overcome them, in a clearer way in abstract and introduction.

(4) More explanation is needed for where there is a research gap and what the goals of the research are. The research gap and the goals of the research are not explained in detail which leads to the reader missing the significance of the research

(5）Inadequate methodology rationale: Justification for the model design and method selection was insufficient and merits elaboration

(6)This article has obtained some interesting findings through the models, but these findings need to be further verified from theory or actual conditions. Also, further highlight the contribution of this article.

(7) The article lacks an important discussion link, in which the author should focus on describing the differences between the article study and other scholars' studies, thus highlighting the relevance and academic value of the article, the following literature should be helpful for your research：(1) Reduction pathways identification of Agricultural Water Pollution in Hubei Province, China. (2)Coordination of the Industrial-Ecological Economy in the Yangtze River Economic Belt, China.

(8) Ambiguous conclusions: Conclusions were somewhat vague; more in-depth synthesis to derive more definitive conclusions would be advantageous

6. PLOS authors have the option to publish the peer review history of their article (what does this mean?). If published, this will include your full peer review and any attached files.

Reviewer #1: No

Reviewer #2: No

---

## [Author Response · Author response to Decision Letter 0]

26 Dec 2023

Thank you for your letter and for the reviewers' and editors' comments concerning our manuscript. These comments are very helpful for revising and improving our paper. We have studied comments carefully and have made corrections which we hope meet with approval.

Reviewer #1comments1: 

The introduction needs to be improved, and it is necessary to expose the innovations that this study brings compared with other urbanization or air pollution studies. We also encourage the authors to add some references to improve the introduction according to the research gaps and the needs of the study. 

[1] Cao N, Luo N, Peng Y, et al. Evaluation of Urbanization Development Dynamics Based on Quadrilateral Diamond Structure Model: Evidence from Chongqing, China, in the New Era[J]. Journal of urban planning and development, 2023,149(3). https://doi.org/10.1061/JUPDDM.UPENG-4413

[2] Sun P, Luo N. Comparative Analysis on the Quality of Urbanization Structure of Central Cities in Southwest Economic Core Area of China: Taking Chengdu-Chongqing as Examples[J]. Scientia Geographica Sinica, 2021,41(6): 1029-1039. DOI:10.13249/j.cnki.sgs.2021.06.011

Response: Thanks for your valuable comments and suggestions. We have improved the introduction by adding innovations brought by this study compared to urbanization or air pollution studies. We have improved the introduction by adding innovations brought by this study compared to urbanization or air pollution studies. At the same time, the following references were added to supplement the introduction based on research gaps and research needs. The details are as follows:

Urbanization is the process of economic, social, land, and population transformation from countryside to towns in the course of a country's development. Among them, the core of urbanization is population urbanization, and the carrier of urbanization is land urbanization [11]. With the intensification of atmospheric environmental pollution, scholars from all walks of life have studied the effect of pollution reduction and carbon reduction from multiple perspectives and found that urbanization is an important factor affecting the atmospheric environment [12]. "China's New Urbanization Report 2012" points out that the non-intensive and crude production of China's urbanization focuses on the quantity and scale of urban development, which ignores the value of environmental resources and is an important cause of the decline in the quality of the atmospheric environment. The problem of air pollution in China has changed from single pollutant pollution to compound pollution, which is increasing with the rapid development of urbanization in China [13] [14]. Civerolo et al. [15] used the SLEUTH model to extrapolate New York City's land cover data from 1990 to the next 2050 and predicted that future urbanization in New York will lead to elevated ozone concentrations. Scholars agree with the conclusion that urbanization has an impact on air pollution. Carbon emissions and haze are the main control gases. Scholars from all walks of life have done a lot of research on them, and the methods are diverse. From the existing literature, the STIRPAT model [16], Dubin model [17], DPSIR model [18], the geographically weighted regression model (GWR) [19], geographical detector [20], and so on are mainly used to study the development law of air pollution. Chameides [21] believed that the urban heat island effect caused by urbanization is the main reason for the increase in urban ozone concentration. Many scholars use the WRF Chem model to check ozone pollution, and it shows that urbanization increases ozone concentration. [22-23]. Overall, the existing research results are relatively rich and provide an empirical basis for exploring the relationship between urbanization and air pollution. However, the exploration of urbanization on ozone pollution is still relatively small. In recent years, the causes of atmospheric pollution have strengthened from a purely environmental problem to a complex ecosystem problem of population, land, economy, and society. Current urbanization studies mostly use land cover data to select indicators and fail to consider other factors that influence ozone concentration. There is also limited research on the atmospheric environment effect from the perspective of urbanization. Therefore, understanding the characteristics of spatial and temporal changes in ozone and the driving mechanisms from the perspective of urbanization is essential to further promote the harmonious development of nature and society.

Considering all the above explanations, the present study proposes to address the following questions: i) The distribution characteristics of ozone in time and space. ii) Under the background of urbanization development, the driving mechanism of ozone pollution is analyzed (from the four urbanization development directions of economy, population, land, and society). iii) To unify the distribution of factors affecting ozone concentration by urbanization development in all provinces of the country. In addition, the distribution and driving mechanisms of ozone pollution from the perspective of China's urbanization are less frequently studied. To answer these questions above, we attempt to use ozone concentration data and statistical yearbook data of 31 provinces in China to study the driving mechanism of ozone pollution under the context of urbanization. We intensify on the impact of urbanization indicators on ozone and analyze the geographical spatial distribution of urbanization impact factors on the ozone impact index. By citing urbanization, hope to be more comprehensive on the causes of China's distribution characteristics of ozone pollution and how to solve this problem are discussed, and thus draw more accurate results and set forward more constructive suggestions. The intensification of atmospheric environmental pollution in China has attracted extensive attention from the government and academia, but the domestic existing literature, focuses on exploring the micro-channels of urbanization on ozone pollution, and the index selection is based on urban land cover data. The research on the impact of other influencing factors of urbanization on ozone concentration is indeed more insufficient. Therefore, this study aims to empirically analyze the impact mechanism and driving factors of ozone pollution in the context of urbanization in China using the dynamic spatial Durbin model and the spatio-temporal geographically weighted regression model. Producing a new explanation for the causes of the spatial and temporal distribution characteristics of ozone pollution, producing a basis for regional ozone and pollution prevention and control measures, has important theoretical and practical relevance for the government to formulate and improve atmospheric environmental protection policies, and how the government will carry out macro in the future.

Thanks! 

Reviewer #1comments2: 

Please ensure the accuracy of data sources: the author uses air pollution data at the prefecture-level city, so the process of aggregating such data to the provincial level requires further explanation. Also, the URL in line 197 contains errors, please check and correct it.

Response: Thanks for your valuable comments and suggestions. We rewrote the data source and processing to correct the URL in line 197. The details are as follows:

Data Sources

Based on the inter-provincial panel data from 2015 to 2019, this paper takes 31 provinces and regions in China as research samples. According to the historical data of 268 prefecture-level cities released by China Air Quality Online Monitoring and Analysis Platform (https://www.aqistudy.cn/historydata/), the concentration data of O3, NO2, CO, SO2, PM10, PM2.5 and AQI in each prefecture-level city were integrated at the provincial level, and the annual average value was used to calculate the concentration data of air pollutants in each province. O3 (8h) is the ozone concentration data obtained by 8-hour ozone moving average concentration detection. We derived the data of urbanization variables and control variables from the National Bureau of Statistics (https://data.stats.gov.cn/easyquery.htm?cn=C01), and we derived all other relevant data from the statistical yearbooks of Chinese provinces.

Thanks!

Reviewer #1comments3:

"Fig 1. Flow chart of urbanization ozone pollution" needs correction. The relationships and connections between different components in Fig. 1 are not clearly characterized. I suggest the author further refine the contents in this figure, especially by establishing clearer links between aspects of urbanization and specific variables (e.g. industrial structure, urbanization rate). This could better explain the rationale behind selecting the independent variables, and illustrate the mechanism linking urbanization and ozone pollution.

Response: Thanks for your valuable comments and suggestions. The specific variable indicators included in the demographic urbanization, land urbanization, social urbanization, economic urbanization and control variables are indicated by using the form of arrows and the same shape of the outer frame of the variable. The details are as follows:

Fig 1. Flow chart of urbanization ozone pollution

Thanks!

Reviewer #1comments4:

The "Characterization of ozone spatio-temporal evolution" in the Empirical Results section does not combine time and space in the analysis, making the contents and results less clear. I suggest the author reorganize this part, for example, combining Table 3 and Fig. 3 to present the results on a map, which could better demonstrate the characteristics of different clustering types.

Response: Thanks for your valuable comments and suggestions. We redrew a map of the spatial distribution of ozone (Figure 3), which can better demonstrate the characteristics of different clustering types. And we use a combination of time and space for the analysis. The details are as follows:

The results of Figure 3 further show that the distribution characteristics of near-surface ozone in China are "east-heavy, west-light," "short in the south and high in the north". The spatial distribution trend of diffusion to central China and coastal cities is more prominent in the Beijing, Tianjin, Hebei, and Shandong provinces. Significant spatial heterogeneity characterizes the inter-province. Hebei Province, Tianjin City, and Shandong Province are the key areas of the petrochemical and organic chemical industry in China, and the stock of coal-fired boilers is relatively large. The region has dense traffic, developed logistics, and huge emissions of diesel trucks and fuel vehicles. These factors have expanded VOCs and nitrogen oxide emissions in these areas, forming a medium and long-term pressure on ozone prevention and control in the region. In 2015, the high-value areas of ozone concentration in China were distributed across the eastern region, mainly in Beijing, Shanghai, Shandong, Jiangsu, Qinghai, Zhejiang, Gansu, and Inner Mongolia. In Shanghai, the average annual concentration reached 105μg/m3. By 2019, the concentration of ozone in China increased compared with 2015, and ozone pollution expanded rapidly in the east, mainly concentrated in North China and East China. Among them, Hebei Province, Tianjin City, and Shanxi Province in North China are the high-value areas of ozone pollution concentration growth in China. The near-surface ozone concentration in Tianjin increased from 77 μg/m3 in 2015 to 106 μg/m3 in 2019. This is because more developed industries, dense populations, and higher emissions of ozone precursors from motor vehicles and industrial sources typically characterize these areas [5].

Fig 3. The spatial distribution of ozone concentration in China's provinces in 2015 and 2019

Thanks!

Reviewer #1comments5:

In the Response Results section, please add validation results of the spatial econometric models (in a table form); also, only very few indicators in Table 4 pass the significance test, and the Coefficients and t-values are inconsistent in signs, please double check and correct these issues.

Response: Thanks for your valuable comments and suggestions. We have added a table of LM test results to the Response Results section and corrected the contents of Table 5. The details are as follows:

Response Results

In order to determine the specific form of the model, this paper carried out the LM test, LR test, and Hausman test. According to the LM results in Table 3, LM rejects the null hypothesis, and the results tend to be the spatial error model (SEM). Further, the LR test is used to verify the model, and it is found that the null hypothesis is rejected at the 1% level. The Hausman test showed that the null hypothesis was rejected at the 1% level. Therefore, the fixed effect spatial error model (SEM) was finally selected for the experiment.

Table 3. LM test of spatial panel model

LM test

 Statistic df p-value

Spatial error 

Moran’s I 113.175 1 0.000

Lagrange multiplier 8.510 1 0.004

Robuet Lagrange multiplier 8.680 1 0.003

Spatial lag 

Lagrange multiplier 0.826 1 0.364

Robuet Lagrange multiplier 0.995 1 0.319

Table 4. Spatial error model under nested matrix

Variable Direct effect Indirect effect Gross effect

 coefficient t coefficient t coefficient t

lngdp 0.082 1.14 0.84 1.28 0.92 1.33

lnind 0.24** 2.56 1.31*** 3.16 1.54*** 3.56

lnpop -0.04 -0.98 0.94* 1.68 0.91 1.56

lnurb 0.10 1.31 28.09** 2.30 29.09** 2.31

lnurb2 0.83* 1.93 16.34** 1.99 17.17** 2.03

lnare 0.03 0.52 0.02 -0.05 0.06 0.12

lnstu -0.10 -0.65 -6.11** -2.33 -6.22** -2.31

lnrd 0.04 1.08 -0.13 -0.30 -0.08 -0.19

Note:* p < 0.1, ** p < 0.05, *** p < 0.01.

Thanks!

Reviewer #1comments6:

Suggest adding policy implications in the Discussion or Conclusion section to highlight the practical significance of this study, and echo the problem the author aims to address - "to provide a reference for the government to formulate and improve atmospheric environmental protection policies".

Response: Thanks for your valuable comments and suggestions. We have added policy recommendations in the conclusion section. The details are as follows:

Conclusion and policy suggestion

After analyzing the distribution and spatial and temporal characteristics of ozone pollution in China, this paper explores the driving mechanism of ozone pollution in the context of urbanization. Using the spatial econometric model and quantitative method, this paper analyzes the regional differences, temporal and spatial distribution patterns, influencing factors, and spatial spillover effects of ozone pollution in China, with the conclusions as follows:

First, the lowest value of ozone concentration in China's ambient air from 2015 to 2019 was in 2015, and the average annual increase was 1.68㎍/m3. There is a significant spatial autocorrelation in the level of ozone pollution. Among them, Beijing, Shanghai, Tianjin, Shandong, Henan, Hebei, Inner Mongolia, Anhui, Liaoning and Jiangsu are always the high value areas of ozone concentration, and the spatial distribution of the agglomeration area in the eastern expansion of the "east-heavy and west-light" and "south-short and north-high" characteristics, which indicates that ozone pollution has obvious regional and agglomeration characteristics.

Second, selecting a spatial econometric model to quantitatively identify the key factors influencing ozone concentration in ambient air, we found that stu and rd have a negative effect on the increase of ozone concentration, while ind, urb, gdp and are have a positive effect on the increase of ozone concentration. Studies have shown that there are not only direct effects that affect the level of local ozone pollution, but also indirect effects that affect the level of adjacent ozone pollution also known as spillover effects.

Last, there is also spatial heterogeneity in the factors influencing the increase in ozone concentration. The regression model analysis shows that the inhibitory effect of stu and rd on the increase of ozone concentration has been reduced, and the promotion effect of urb rate, are and gdp on the increase of ozone concentration has accumulate

---

## [Decision Letter · Decision Letter 1]

8 Jan 2024

PONE-D-23-35953R1Spatialtemporal evolution characteristics of ozone in China and its response to urbanizationPLOS ONE

Dear Dr. Wu,

Thank you for submitting your manuscript to PLOS ONE. After careful consideration, we feel that it has merit but does not fully meet PLOS ONE’s publication criteria as it currently stands. Therefore, we invite you to submit a revised version of the manuscript that addresses the points raised during the review process.

We look forward to receiving your revised manuscript.

Kind regards,

Shazia Rehman, Ph.D.

Academic Editor

PLOS ONE

Journal Requirements:

Reviewers' comments:

Reviewer's Responses to Questions

**Comments to the Author**

1. If the authors have adequately addressed your comments raised in a previous round of review and you feel that this manuscript is now acceptable for publication, you may indicate that here to bypass the “Comments to the Author” section, enter your conflict of interest statement in the “Confidential to Editor” section, and submit your "Accept" recommendation.

Reviewer #1: All comments have been addressed

Reviewer #2: (No Response)

2. Is the manuscript technically sound, and do the data support the conclusions?

Reviewer #1: Yes

Reviewer #2: Yes

3. Has the statistical analysis been performed appropriately and rigorously? 

Reviewer #1: Yes

Reviewer #2: Yes

4. Have the authors made all data underlying the findings in their manuscript fully available?

Reviewer #1: Yes

Reviewer #2: Yes

5. Is the manuscript presented in an intelligible fashion and written in standard English?

Reviewer #1: Yes

Reviewer #2: Yes

6. Review Comments to the Author

Reviewer #1: The manuscript has been revised in accordance with the comments and is essentially ready for publication. It is hoped that the authors will further revise the formatting of the manuscript (e.g., clarity of figures, formatting of references, etc.) in accordance with the requirements of the journal.

References 41-43 are not labeled in the manuscript, please revise it.

Reviewer #2: (1) The title of the article needs to be revised.

(2) The innovation of this paper needs to be highlighted in the abstract, and the background of the paper needs to modify.

(3) The literature review is not enough, the contribution made by previous studies has not been clearly expressed, and the author needs to introduce the research methods of previous articles to highlight the innovation of this paper

(4）The article was not written following the correct journal's guidelines to be considered for publication. INTORDCUTION→MRTHOD→RESULTS→DISSCUSION→CONCLUSION

(5) The discussion should focus on describing the differences between the article study and other scholars' studies, thus highlighting the relevance and academic value of the article, the following literature should be helpful for your research：(1) A Set Pair Analysis Method for Assessing and Forecasting Water Conflict Risk in Transboundary River Basins, (2)Reduction pathways identification of Agricultural Water Pollution in Hubei Province, China.

(6) The format of the article should be standardized, and the quality of the figures and tables needs to be improved.

7. PLOS authors have the option to publish the peer review history of their article (what does this mean?). If published, this will include your full peer review and any attached files.

Reviewer #1: **Yes: **PINGJUN SUN

Reviewer #2: No

---

## [Author Response · Author response to Decision Letter 1]

20 Feb 2024

Thank you for your letter and for the reviewers' and editors' comments concerning our manuscript. These comments are very helpful for revising and improving our paper. We have studied comments carefully and have made corrections which we hope meet with approval.

Reviewer #1comments1: 

The manuscript has been revised in accordance with the comments and is essentially ready for publication. It is hoped that the authors will further revise the formatting of the manuscript (e.g., clarity of figures, formatting of references, etc.) in accordance with the requirements of the journal. References 41-43 are not labeled in the manuscript, please revise it.

Response: Thanks for your valuable comments and suggestions. We have re-formatted the manuscript as required by the journal and have also made changes to the references. The specific changes are reflected in the manuscript.

Thank you very much for your valuable advice!

Reviewer #2comments1: 

The title of the article needs to be revised.

Response: Thanks for your valuable comments and suggestions. The present study proposes to address the following questions: i) The distribution characteristics of ozone in time and space. ii) Under the background of urbanization development, the driving mechanism of ozone pollution is analyzed (from the four urbanization development directions of economy, population, land, and society). iii) To unify the distribution of factors affecting ozone concentration by urbanization development in all provinces of the country. To answer these questions above, we attempt to use ozone concentration data and statistical yearbook data of 31 provinces in China to study the driving mechanism of ozone pollution under the context of urbanization. Exploring the response of ozone pollution to urbanization.

In summary, we believe that this title "Spatialtemporal evolution characteristics of ozone in China and its response to urbanization" is consistent with the main thrust of the manuscript's.

Thanks!

Reviewer #2comments2: 

The innovation of this paper needs to be highlighted in the abstract, and the background of the paper needs to modify.

Response: Thanks for your valuable comments and suggestions. We have reworked and rewritten the abstract to add the significance and innovation of the article. The details are as follows:

Abstract

Based on the background of urbanization in China, we used the dynamic spatial panel Durbin model to study the driving mechanism of ozone pollution empirically. We also analyzed the spatial distribution of ozone driving factors using the GTWR. The results show that: i) The average annual increase of ozone concentration in ambient air in China from 2015 to 2019 was 1.68μg/m3, and 8.39μg/m3 elevated the year 2019 compared with 2015. ii) The Moran's I value of ozone in ambient air was 0.027 in 2015 and 0.209 in 2019, showing the spatial distribution characteristics of "east heavy and west light" and "south low and north high". iii) Per capita GDP industrial structure, population density, land expansion, and urbanization rate have significant spillover effects on ozone concentration, and the regional spillover effect is greater than the local effect. R&D intensity and education level have a significant negative impact on ozone concentration. iv) There is a decreasing trend in the inhibitory effect of educational attainment and R&D intensity on ozone concentration, and an increasing trend in the promotional effect of population urbanization rate, land expansion, and economic development on ozone concentration. Empirical results suggest a twofold policy meaning: i) to explore the causes behind the distribution of ozone from the new perspective of urbanization, and to further the atmospheric environmental protection system and ii) to eliminate the adverse impacts of ozone pollution on nature and harmonious social development.

Thanks!

Reviewer #2comments3: 

The literature review is not enough, the contribution made by previous studies has not been clearly expressed, and the author needs to introduce the research methods of previous articles to highlight the innovation of this paper.

Response: Thanks for your valuable comments and suggestions. We have reworked and supplemented the literature review section to summarise the relevant research methods of scholars and highlight the innovative nature of this paper. The details are as follows:

Urbanization is the process of economic, social, land, and population transformation from countryside to towns in the course of a country's development. Among them, the core of urbanization is population urbanization, and the carrier of urbanization is land urbanization [11]. With the intensification of atmospheric environmental pollution, scholars from all walks of life have studied the effect of pollution reduction and carbon reduction from multiple perspectives and found that urbanization is an important factor affecting the atmospheric environment [12]. "China's New Urbanization Report 2012" points out that the non-intensive and crude production of China's urbanization focuses on the quantity and scale of urban development, which ignores the value of environmental resources and is an important cause of the decline in the quality of the atmospheric environment. The problem of air pollution in China has changed from single pollutant pollution to compound pollution, which is increasing with the rapid development of urbanization in China [13-14]. Civerolo et al. [15] used the SLEUTH model to extrapolate New York City's land cover data from 1990 to the next 2050 and predicted that future urbanization in New York will lead to elevated ozone concentrations. Scholars agree with the conclusion that urbanization has an impact on air pollution. Carbon emissions and haze are the main control gases. Scholars from all walks of life have done a lot of research on them, and the methods are diverse. From the existing literature, the STIRPAT model [16], Dubin model [17], DPSIR model [18], the geographically weighted regression model (GWR) [19], geographical detector [20], and so on are mainly used to study the development law of air pollution. The Theil coefficient, Theil coefficient, coefficient of variation, and spatial autocorrelation (global Moran's I index and local G coefficient) are often used to reveal the regional differences and spatial correlation of air pollution intensity. For example, Sun Yaohua et al. [21] explored the differences in carbon emission intensity between provinces and regions in China-based on the Theil index and found that there were differences in carbon emission intensity between provinces in China and accumulated. Zhao Yuntai et al. [22] divided the entire country into eight economic regions and used the Theil index, global autocorrelation Moran's I, and cold and hot spot analysis methods to explore the spatial evolution characteristics of regional carbon emission intensity. It was found that the widening difference between regions caused the widening difference of regional carbon emission intensity in the entire country, while the difference within the region was small. The research on the relationship between urbanization and air pollution mainly forms three conclusions: the positive linear relationship between urbanization and air pollution [23-24], the negative linear relationship between urbanization and air pollution [25], and the nonlinear curve relationship between urbanization and air pollution [26]. Among them, scholars have conducted empirical investigations on the different influencing factors of ozone on air pollution, but there are few studies on the impact of ozone distribution in air pollution from the perspective of urbanization.

Thanks!

Reviewer #2comments4: 

The article was not written following the correct journal's guidelines to be considered for publication. INTORDCUTION→MRTHOD→RESULTS→DISSCUSION→CONCLUSION.

Response: Thanks for your valuable comments and suggestions. We have followed the correct journal guidelines for rewriting. The structure is: 

INTORDCUTION→MATERIAL AND METHODS→RESULTS→DISCUSSION→CONCLUSION.

Thanks!

Reviewer #2comments5:

The discussion should focus on describing the differences between the article study and other scholars' studies, thus highlighting the relevance and academic value of the article, the following literature should be helpful for your research：(1) A Set Pair Analysis Method for Assessing and Forecasting Water Conflict Risk in Transboundary River Basins, (2)Reduction pathways identification of Agricultural Water Pollution in Hubei Province, China.

Response: Thanks for your valuable comments and suggestions. We refer to the methodology of these two papers, the introductory section and the main content of the study changed the introductory section and added to the references of this paper. The details are as follows: 

Urbanization is the process of economic, social, land, and population transformation from countryside to towns in the course of a country's development. Among them, the core of urbanization is population urbanization, and the carrier of urbanization is land urbanization [11]. With the intensification of atmospheric environmental pollution, scholars from all walks of life have studied the effect of pollution reduction and carbon reduction from multiple perspectives and found that urbanization is an important factor affecting the atmospheric environment [12]. "China's New Urbanization Report 2012" points out that the non-intensive and crude production of China's urbanization focuses on the quantity and scale of urban development, which ignores the value of environmental resources and is an important cause of the decline in the quality of the atmospheric environment. The problem of air pollution in China has changed from single pollutant pollution to compound pollution, which is increasing with the rapid development of urbanization in China [13-14]. Civerolo et al. [15] used the SLEUTH model to extrapolate New York City's land cover data from 1990 to the next 2050 and predicted that future urbanization in New York will lead to elevated ozone concentrations. Scholars agree with the conclusion that urbanization has an impact on air pollution. Carbon emissions and haze are the main control gases. Scholars from all walks of life have done a lot of research on them, and the methods are diverse. From the existing literature, the STIRPAT model [16], Dubin model [17], DPSIR model [18], the geographically weighted regression model (GWR) [19], geographical detector [20], and so on are mainly used to study the development law of air pollution. The Theil coefficient, Theil coefficient, coefficient of variation, and spatial autocorrelation (global Moran's I index and local G coefficient) are often used to reveal the regional differences and spatial correlation of air pollution intensity. For example, Sun Yaohua et al. [21] explored the differences in carbon emission intensity between provinces and regions in China-based on the Theil index and found that there were differences in carbon emission intensity between provinces in China and accumulated. Zhao Yuntai et al. [22] divided the entire country into eight economic regions and used the Theil index, global autocorrelation Moran's I, and cold and hot spot analysis methods to explore the spatial evolution characteristics of regional carbon emission intensity. It was found that the widening difference between regions caused the widening difference of regional carbon emission intensity in the entire country, while the difference within the region was small. The research on the relationship between urbanization and air pollution mainly forms three conclusions: the positive linear relationship between urbanization and air pollution [23-24], the negative linear relationship between urbanization and air pollution [25], and the nonlinear curve relationship between urbanization and air pollution [26]. Among them, scholars have conducted empirical investigations on the different influencing factors of ozone on air pollution, but there are few studies on the impact of ozone distribution in air pollution from the perspective of urbanization.

References：

1. Wei Zhao, Bo Gao, Qing Lu, Zhiqiang Zhong, Xiaoming Liang, Ming Liu, Shexia Ma, Jiaren Sun, Laiguo Chen, Shaojia Fan. Ozone pollution trend in Pearl River Delta region from 2006 to 2019. Environmental Science. 2021; 42(1), 97-105.

2. Zhongling Hu. Ozone pollution in China is increasing year by year. Ecological Economy. 2020; 36(9), 5-8.

3. Zhenhua Zhang, Jing Wang, Yanchao Feng, Wenjia Tian. Effect of green finance reform and innovation pilot zone on ozone pollution. Chinese Mouth Resources and Environment. 2022; 32(12), 52-65.

4. Shutao Chen, Yong Zhang, Zhenhua Hu, Yanshu Shi, Xiaoshuai Shen. Effects of elevated ozone concentration and soil moisture on respiration temperature sensitivity of soil microorganisms in farmland. Environmental Science. 2012; 33(05), 1476-1483.

5. Tianyue Zhang, Nanchi Shen, Xue Zhao, Xinyu Wang, Wenji Zhao. Spatial-temporal variation characteristics of ozone concentration and population exposure risk assessment in Chengdu-Chongqing urban agglomeration from 2015 to 2019. Journal of Environmental Science. 2021; 41(10), 4188-4199.

6. Weipeng Ye, Miaomiao Liu, Jun Bi. A meta-analysis of the relationship between short-term ozone exposure and human mortality in China. Journal of Environmental Science. 2020; 40(7), 2644-2651.

7. Weiwu Wang, Chao Chen. Quantitative analysis of spatial distribution and influencing factors of urban air pollutants in Hangzhou. Geographical Research. 2008; (2), 241-249+481.

8. Ministry of Ecology and Environment of the People's Republic of China Report on air quality improvement in China (2013-2018). Beijing:Ministry of Ecology and Environment. 2019.

9. Hua Jiang, Jian Gao, Hong LI, Wanghui Chu, Fahe Chai. A preliminary study on the theoretical framework of coordinated prevention and control of air pollution in China. Research of Environmental Sciences. 2022; 35(3), 601-610.

10. Xiaozhe Wang, Sha Zhao, Linghui Guo, Hebing Zhang, Jiangbo Gao. Seasonal variation of ozone in "2+26" cities in Beijing-Tianjin-Hebei region and surrounding areas. Research of Environmental Sciences. 2022; 35(8), 1786-1797.

11. Xi Yang, Xinhai Lu. Driving factors of urban land urbanization in China from the perspective of spatial effects. Chinese Mouth Resources and Environment. 2021; 31(1), 156-164.

12. Jianneng Xiao, Guoming Du, Yiqiang Shi, Youyue Wen, Yao Jie. Spatial-temporal characteristics of ambient air pollution in Xiamen and its correlation with meteorological factors. Journal of Environmental Science. 2016; 36(9), 3363-3371.

13. Wenyuan Niu. Report on New Urbanization in China·2012, Science Press. ISBN: 9787030356246. 2012.

14. Lijian Han, Weiqi Zhou, Steward T P, Weifeng Li, Yuguo Qian. Multicontaminant air pollution in Chinese cities. Bulletin of the World Health Organization. 2018; 96(4), 233. https://doi.org/10.2471/blt.17.195560

15. Civerolo K, Hogrefe C, Lynn B, Rosenthal J, Ku J Y, Solecki W, Kinney P. Estimating the effects of increased urbanization on surface meteorology and ozone concentrations in the New York City metropolitan region. Atmospheric environment. 2007; 41(9), 1803-1818. https://doi.org/10.1016/j.atmosenv.2006.10.076

16. Rui Huang, Zheng Wang, Guanqun Ding, Yangran Gong, Changxin Liu. Analysis and trend prediction of influencing factors of energy consumption carbon emissions in Jiangsu Province based on STIRPAT model. Geographical Research. 2016; 35(4), 781-789.

17. Shaojian Wang, Jingyuan Zeng, Yongyuan Huang, Chenyi Shi, Peiyu Zhan. The effects of urbanization on CO2 emissions in the Pearl River Delta: a comprehensive assessment and panel data analysis. Applied Energy. 2018; 228, 1693-1706. https://doi.org/10.1016/j.apenergy.2018.06.155

18. Qinlin Xiao, Chao Tian, Yanjun Wang, Xiuqing Li, Liming Xiao. Measurement and comparison of urban haze control level and efficiency based on DPSIR model: A case study of 31 cities in North China. Journal of Resources and Ecology. 2020; 11(6), 549-561.

19. Bin Zou, Xin Fang, Huihui Feng, Xiang Zhou. Simplicity versus accuracy for estimation of the PM2.5 concentration: A comparison between LUR and GWR methods across time scales. Journal of Spatial Science.

---

## [Editor Report · Decision Letter 2]

23 Feb 2024

Spatialtemporal evolution characteristics of ozone in China and its response to urbanization

PONE-D-23-35953R2

Dear Dr. Xiang-Li Wu,

We’re pleased to inform you that your manuscript has been judged scientifically suitable for publication and will be formally accepted for publication once it meets all outstanding technical requirements.

Kind regards,

Shazia Rehman, Ph.D.

Academic Editor

PLOS ONE
---

## [Editor Report · Acceptance letter]

30 Apr 2024

PONE-D-23-35953R2 

PLOS ONE

Dear Dr. Wu, 

I'm pleased to inform you that your manuscript has been deemed suitable for publication in PLOS ONE. Congratulations! Your manuscript is now being handed over to our production team.

Kind regards, 

on behalf of

Dr. Shazia Rehman 

Academic Editor

PLOS ONE